

# Strengthening the atom-field coupling through the deep-strong regime via pseudo-Hermitian Hamiltonians

M. A. de Ponte[1], F. de Oliveira Neto[2], P. M. Soares[2] and M. H. Y. Moussa[2]

**1** Universidade Estadual Paulista (UNESP), Campus Experimental de Itapeva,
18409-010, Itapeva, São Paulo, Brazil
**2** Instituto de Física de São Carlos, Universidade de São Paulo,
P.O. Box 369, São Carlos, 13560-970, SP, Brazil

## Abstract

We present a strategy for strengthening the atom-field interaction through a pseudo-Hermitian Jaynes-Cummings Hamiltonian. Apart from the engineering of an effective non-Hermitian Hamiltonian, our strategy also relies on the accomplishment of short-time measurements on canonically conjugate variables. The resulting fast Rabi oscillations may be used for many quantum optics purposes and specially to shorten the processing time of quantum information.



## 1 Introduction

At the beginning of the 1990s we witnessed remarkable developments in platforms of matter-field interactions [1, 2], allowing the manipulation of the interplay between the matter and field block buildings. Essentially, these were due the achieved intensity of the matter-field coupling compared to the lifetimes of the involved electronic and field states. Concomitantly,

there has been a breakthrough in the field of quantum computation and communication [3] triggered by the quantum algorithm for the factorization of integers presented by P. Shor [4].

The symbiosis between the theoretical proposition of schemes for the implementation of quantum logical operations and their practical realization through the advances achieved in the area of mater-field interaction in the early 1990s, grounded the quantum information theory, giving this subject the current status of an independent and most prominent research area. In addition to experimentally established proofs-of-principles for quantum information processing [1–3], the foundations on quantum mechanics [1,2,5] also benefited greatly from the dialogue between theory and experimentation that spread from the physics of matter-field interaction to nuclear magnetic resonance, cold atoms, and solid-state physics.

Apart from the computational gain afforded by quantum qubits and algorithms, it is the goal of the present work to investigate, in the domain of matter-field interaction, the possibility of further increasing this gain by strengthening the hitherto achieved matter-field coupling. This strengthening should result in a shorter time for excitation exchange between matter and field, and then for quantum information processing. To attain it, we turn to another major advance that occurred in the late 1990s: The quantum mechanics of $\mathcal{PT}$-symmetric Hamiltonians [6,7]. Similarly to what happened with quantum information, the pseudo-Hermitian quantum mechanics is currently an independent research field benefiting from strong activity and interesting results [8].

We remark that the possibility of achieving faster than Hermitian quantum mechanics was long envisioned in Ref. [9]. The challenge then posed is the quantum brachistochrone problem: the search for a Hamiltonian who governs the evolution of a given initial state to a given final state in the least time interval $\tau$. The authors concluded that for Hermitian Hamiltonians $\tau$ has a nonzero lowerbound, whereas for pseudo-Hermitian Hamiltonians it can be made arbitrarily small. However, in contradiction to this remarkable conclusion, it was subsequently found [10] that an inconsistency in the method proposed in [9] actually prevents it from achieving faster than Hermitian evolutions..The protocol we present here is an alternative to achieve faster than Hermitian evolutions by strengthening the atom-field coupling through pseudo-Hermitian interactions. Furthermore, strengthening the atom-field coupling presents a wide range of practical applications in quantum optics [11].

## 2 The effective pseudo-Hermitian Hamiltonian

Our scheme for enhancing the atom-field coupling begins with the construction of an effective non-Hermitian Hamiltonian $H_{eff}$, from the fundamental Jaynes-Cummings (JC) interaction ($\hbar = 1$)

$$H = \lambda \left( a\sigma_+ + a^\dagger \sigma_- \right) , \tag{1}$$

where $\lambda$ is the well-known Rabi frequency, the field ($\omega a^\dagger a$), of frequency $\omega$, is described by the creation and the annihilation operators $a^\dagger$ and $a$, and the two-level atom ($\omega_0 \sigma_z/2$), with frequency $\omega_0$ and excited and ground states $e$ and $g$, is described by the pseudo-spin operators $\sigma_z = |e\rangle\langle e| - |g\rangle\langle g|$, $\sigma_+ = |e\rangle\langle g|$ and $\sigma_- = |g\rangle\langle e|$. The engineering of the effective interaction is one of the main challenges of our protocol, and we address it through the method of the adiabatic elimination of fast variables. For now we start from the premise of an effective non-Hermitian Hamiltonian for the atom-field interaction:

$$H_{eff} = \lambda \left( \alpha a\sigma_+ + \beta a^\dagger \sigma_- \right) , \tag{2}$$

where $\alpha$ and $\beta$ are assumed to be real and positive dimensionless parameters defined in the range $[0,1]$ for a second-order effective interaction $H_{eff}$.

For treating the non-Hermitian Hamiltonian we follow the procedure in Ref. [7], by constructing an Hermitian counterpart of $H_{eff}$ through a nonunitary Dyson map $\eta$, i.e,

$$h = \eta H_{eff} \eta^{-1}. \tag{3}$$

The pseudo-Hermiticity relation $\Theta H_{eff} = H_{eff}^{\dagger} \Theta$ ensures $h = h^{\dagger}$, and the metric operator $\Theta = \eta^{\dagger} \eta$ ensures the unitarity of the time-evolution of the state vector of the non-Hermitian $H_{eff}$. In fact, through the map $\eta$, the pseudo-Hermitian $H_{eff}$, governing the Schrödinger equation $i\hbar\partial_t |\Psi(t)\rangle = H_{eff} |\Psi(t)\rangle$, is transformed into its Hermitian counterpart $h$ governing the equation $i\hbar\partial_t |\psi(t)\rangle = h |\psi(t)\rangle$, where $|\Psi(t)\rangle = \eta^{-1} |\psi(t)\rangle$. In the metric defined by operator $\Theta = \eta^{\dagger}\eta$, it is straightforward to verify the unitarity of the time-evolution of $|\Psi(t)\rangle$, defined by $\langle\Psi(t)|\Psi(t)\rangle_{\Theta} \equiv \langle\Psi(t)|\Theta|\Psi(t)\rangle = \langle\psi(t)|\psi(t)\rangle$. The computation of the matrix elements of the observables $\mathcal{O} = \eta^{-1} o \eta$ [7,12] associated with $H_{eff}$ are accordingly defined by

$$\left\langle \Psi(t)|\mathcal{O}|\tilde{\Psi}(t)\right\rangle_{\Theta} \equiv \langle\Psi(t)|\Theta\mathcal{O}|\Psi(t)\rangle = \langle\psi(t)|o|\psi(t)\rangle, \tag{4}$$

with $o$ being the observables associated with the Hermitian $h$.

We next outline our protocol starting from the engineered Hamiltonian $H_{eff}$ to construct its pseudo-Hermitian counterpart $h$ through the ansatz for the nonunitary Dyson map

$$\eta = \exp\left[\epsilon\left(a^{\dagger}a + 1/2\right) + \mu a^2 + \nu\left(a^{\dagger}\right)^2\right] \otimes \mathbf{1}, \tag{5}$$

where the parameters $\epsilon$, $\mu$, and $\nu$ are assumed to be real and the identity operator $\mathbf{1}$ stands for the atomic subspace. The operator $\eta$ is a positive non-Hermitian operator for $\epsilon^2 - 4\mu\nu > 0$. Defining $\Lambda_{\pm} = \theta \coth\theta \pm \epsilon$, with $\theta = \sqrt{\epsilon^2 - 4\mu\nu}$, we obtain from Eq. (3) the Hamiltonian

$$h = \lambda \frac{\sinh\theta}{\theta}\left[\alpha\left(\Lambda_- a - 2\nu a^{\dagger}\right)\sigma_+ + \beta\left(2\mu a + \Lambda_+ a^{\dagger}\right)\sigma_-\right]. \tag{6}$$

Assuming $\alpha = |\alpha|e^{i\varphi_{\alpha}}$ and $\beta = |\beta|e^{i\varphi_{\beta}}$ with $\varphi_{\beta} = -\varphi_{\alpha}$, a condition that must be imposed when engineering the Hamiltonian 2, the Hermiticity of $h$ demands the relations

$$\epsilon = sgn(|\alpha| - |\beta|)\frac{\ln\Upsilon}{2\sqrt{1+z^2}}, \tag{7a}$$

$$\mu = sgn(|\alpha| - |\beta|)\frac{z\ln\Upsilon}{4\sqrt{R}\sqrt{1+z^2}}, \tag{7b}$$

$$\nu = -sgn(|\alpha| - |\beta|)\frac{z\sqrt{R}\ln\Upsilon}{4\sqrt{1+z^2}}, \tag{7c}$$

$$\theta = sgn(|\alpha| - |\beta|)\frac{\ln\Upsilon}{2}, \tag{7d}$$

where we have defined the Hermiticity degree

$$R = (|\beta|/|\alpha|)^{sgn(|\alpha|-|\beta|)},$$

such that $0 \leq R \leq 1$ for $|\alpha| > |\beta|$ or $|\alpha| < |\beta|$. The ratio $R$ thus decreases monotonically from unity as the Hamiltonian $H_{eff}$ moves away from Hermiticity. We have also defined the quantity

$$\Upsilon = \frac{1+R+(1-R)\sqrt{1+z^2}}{1+R-(1-R)\sqrt{1+z^2}} = \frac{1}{z_{max}^2 - z^2}\left(\frac{1+R}{2\sqrt{R}}z_{max} + \sqrt{1+z^2}\right)^2, \tag{8}$$

and the only free parameter of the map, the positive real

$$z = \sqrt{-4\mu\nu/\epsilon^2} \leq z_{max}, \tag{9}$$

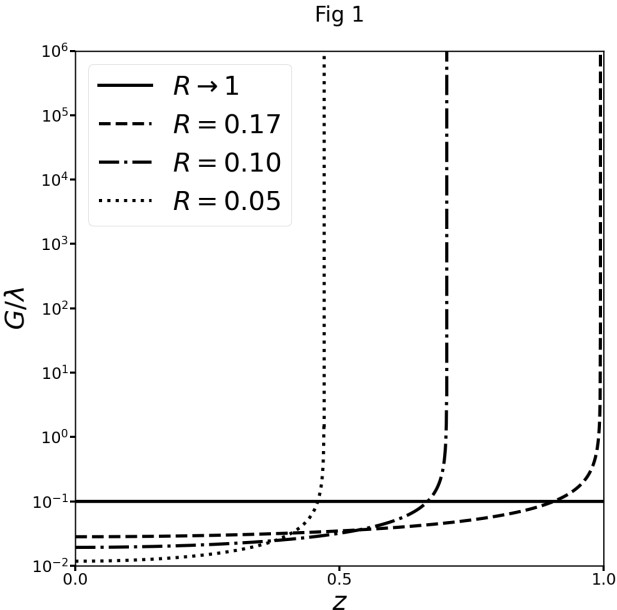

Figure 1: Plot of $G/\lambda$ against $z$ for $R = 1.0, 0.17, 0.1$ and $0.05$, as indicated by solid, dashed, dashed-dotted and dotted lines, respectively, assuming $\alpha = 10^{-1} > \beta$.

which is bounded, for a given $R$, to the maximum $z_{\max} = \min\left[2\sqrt{R}/(1-R), 1\right] \leq 1$, since $z > z_{\max}$ leads to the forbidden $\theta < 0$ as well as $\Upsilon < 0$. For $z_{\max} = 1$ we obtain $R_{\max} \approx 0.17$, showing that the enhancement of the atom-field coupling, prevented for $R > R_{\max}$, demands Hamiltonians with a significantly small degree of Hermiticity. By fixing $R$ and $z$ we automatically obtain $\epsilon$, $\mu$, and $\nu$ from Eq. (7), and defining $\chi = 2z/z_{\max} \leq 2$, such that $0 < \chi \leq 2$, we end up with the Hermitian counterpart of $H_{eff}$:

$$h = G\left[a\sigma_+ + a^\dagger\sigma_- + \chi\left(a^\dagger\sigma_+ + a\sigma_-\right)\right], \tag{10}$$

where the effective coupling strength is given by

$$G = \alpha\lambda\Lambda_-\sinh\theta/\theta. \tag{11}$$

The Rabi frequency $G$ increases proportionally to $\theta$, diverging when $\theta \to \infty$, what happens, for a given $R$, when $1 + R - (1-R)\sqrt{1+z^2} \to 0$ or, equivalently, for $z$ approaching the upper physical limit $z_{\max}$ and $\chi \to 2$. As expected, the counter-rotating terms inevitably contribute when the effective Rabi frequency starts to increase, from the neighborhood of the strong-coupling ($G \approx \omega \approx \omega_0$) through the deep-strong coupling regime ($G \gg \omega \approx \omega_0$). The growth of the effective coupling $G$ relative to the Rabi frequency $\lambda$ implies in a shortened period for the atomic inversion $\langle\sigma_z(t)\rangle$ or excitation exchange, proportional to $1/G$ instead of $1/\lambda$.

In Fig. 1 we plot $G/\lambda$ against $z$ for distinct values of $R = \beta/\alpha$, assuming $\alpha = 10^{-1} > \beta$. The choice of $\alpha$ and $\beta$ smaller than unity is due to the fact that effective Hamiltonians are generally second-order approximations of the original interactions. The solid line follows for $R \to 1.0$, with the respective Hermitian Hamiltonian being a second-order approximation of the original Jaynes-Cummings interaction with a constant coupling $G = \alpha\lambda$, such that $G/\lambda = 10^{-1}$. The dashed line, starting from $G/\lambda = 4.1 \times 10^{-2}$, follows for $R_{\max} \simeq 0.17$. We also consider the dashed-dotted and the dotted lines for $R = 0.1$ and $0.05$, which start from $G/\lambda = 3.2 \times 10^{-2}$ and $2.2 \times 10^{-2}$, respectively. Therefore, when $z$ is sufficiently far from $z_{\max}$, the effective coupling $G$ is around two orders of magnitude smaller than the original Rabi

frequency $\lambda$, increasing slowly before reaching the vicinity of $z_{\max}$ ($= \infty$, 1.0, 0.7, 0.47 for $R = 1.0, 0.17, 0.1, 0.05$, respectively) when it presents an abrupt growth through the strong and deep-strong coupling regimes. The atom-field interaction energy thus grows proportionally to $G$, leading us to conclude that the energy required for engineering $H_{eff}$ must grow as we move away from Hermiticity, decreasing $R$. In other words, the engineerring of $H_{eff}$ for $R \lesssim 0.17$, must demand the action of strong amplification fields to sustain the strength of the atom-field coupling $G$. In short, Fig. 1 shows that we can control the atom-field coupling strength by controlling the Hermiticity degree $R$, at the expense of providing enough energy to engineer the effective interaction $H_{eff}$.

## 3 A cost-benefit analysis: Sensitive issues of our scheme

We next analyze the cost of this extraordinary gain in the atom-field interaction energy, starting with carrying out the necessary measurements on the observables related to the pseudo-Hermitian system $H_{eff}$. These observables are computed from those related to the Hermitian system $h$ through the expression $\mathcal{O} = \eta^{-1} o \eta$. Considering, for example, the quadratures of the radiation field, given by $x_1 = (a + a^{\dagger})/2$ and $x_2 = (a - a^{\dagger})/2i$ for the Hermitian system, we obtain for $H_{eff}$ the transformed observables

$$\mathcal{X}_1 = \eta^{-1} x_1 \eta = \mathcal{A} x_1 + \mathcal{B} x_2 \,, \tag{12a}$$

$$\mathcal{X}_2 = \eta^{-1} x_2 \eta = \mathcal{A} x_2 + \mathcal{B} x_1 \,, \tag{12b}$$

with coefficients $\mathcal{A} = (\theta \coth \theta + \nu - \mu) \sinh \theta / \theta$ and $\mathcal{B} = (\epsilon - \nu - \mu) \sinh \theta / \theta$, both diverging as $z \to z_{\max}$. Therefore, the knowledge of $\mathcal{X}_1$ and $\mathcal{X}_2$ follows from the simultaneous measurements of the canonically conjugated variables $x_1$ and $x_2$, whose accomplishment is discussed in Refs. [16, 17].

Regarding achieving faster than Hermitian quantum mechanics, we note that the effective coupling strength $G$ defines a typical time $1/G$ to carry out an elementary logical operation. The minimum energy required for this operation, over a given error tolerance $\varepsilon$, is estimated to be $E_{min} \approx \hbar G / \varepsilon$ [13]. The higher the Rabi frequencies, the higher the energies required for this fast than Hermitian quantum operation, as higher as the lower the error tolerances.

We mention here a recently presented result [14], where it is demonstrated that the construction of coherent many-body Rabi oscillations, through the coherent interaction of an atomic sample with a field mode, allows increasing the Rabi frequency $g$ by the factor $\sqrt{N}$, where $N$ is the number of atoms in the sample. In this case, the typical time to carry out an elementary logical operation decreases from $1/g$ to $1/\sqrt{N}g$. Therefore, in addition to the gain in computational time that results from the quantum nature of the operation, i.e., from the use of qubits as information carriers [15], we have here the gain that results from the collective nature of the radiation-matter interaction. In the present proposal, the gain in computational time comes from strengthening the Rabi frequency through pseudo-Hermiticity instead of taking advantage of collective effects in the coherent interaction between atomic samples and cavity fields.

### 3.1 The construction of the effective non-Hermitian Hamiltonian

An additional cost for strengthening the atom-field coupling regards the engineering of the non-Hermitian Hamiltonian $H_{eff}$, which must demand a large supply of energy, as large as that made available by the atom-field interaction $G$. Consequently, the usual method of engineering Hamiltonians by the adiabatic elimination of fast variables [18], which requires the amplitudes

Fig 2

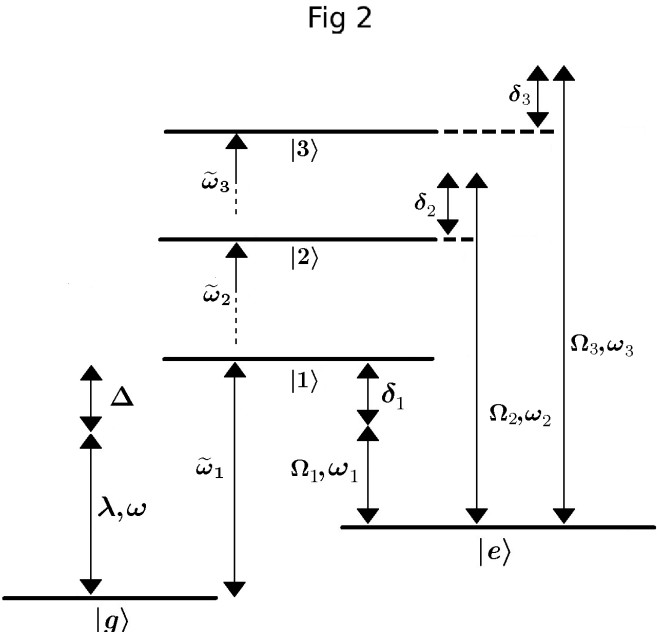

Figure 2: Atomic configuration to engineer the non-Hermitian Jaynes-Cummings interaction.

of the amplification fields to be much smaller than their detunings with the pumped system, should not apply to these cases, as discussed below.

Let us consider the atom-field interactions sketched in Fig. 2, where the ground ($|g\rangle$) and excited ($|e\rangle$) states are coupled through Raman transitions to $N$ auxiliary adjacent states $|1\rangle,...,|N\rangle$, labeled by the frequencies $\tilde{\omega}_\ell$. In Fig. 2 we only show the adjacent levels $|1\rangle$, $|2\rangle$ and $|3\rangle$. A quantum mode $\omega$ and $N$ classical fields $\omega_\ell$ ($\ell = 1,...,N$) are considered for this purpose. The mode is set to drive the transition $|g\rangle \leftrightarrow |1\rangle$ with strength $\lambda$ and detuning $\Delta = \tilde{\omega}_1 - \omega$, while the $\ell$-th classical field is set to drive the transition $|e\rangle \leftrightarrow |\ell\rangle$ with strength $\Omega_\ell$ and detuning $\Delta_\ell = (-1)^{\delta_{\ell 1}}(\omega_e + \omega_\ell - \tilde{\omega}_\ell)$, setting the energy of the ground state $|g\rangle$ to zero. The Hamiltonian describing the process is given by $H = H_0 + V$, with

$$H_0 = \omega a^\dagger a + \omega_e \sigma_{ee} + \sum_\ell \tilde{\omega}_\ell \sigma_{\ell\ell}, \tag{13}$$

$$V = \lambda a \sigma_{1g} + \sum_\ell \Omega_\ell \sigma_{\ell e} e^{-i\omega_\ell t} + H.c., \tag{14}$$

where $a^\dagger$ ($a$) is the creation (annihilation) operator for the mode and $\sigma_{uv} = |u\rangle\langle v|$ represents pseudo-spin operators, with $u, v = e, g, \ell$. Under the conditions $\Delta \gg \sqrt{\bar{n}}|\lambda|$, $\bar{n}$ being the average excitation of the mode, and $\Delta_\ell \gg |\Omega_\ell|$, which imposes severe limitation on the amplitude of the pumping fields, the Hamiltonian in the interaction picture

$$\mathcal{H}(t) = \lambda a \sigma_{1g} e^{i\Delta t} + \sum_\ell \Omega_\ell \sigma_{\ell e} e^{-(-1)^{\delta_{\ell 1}} i\Delta_\ell t} + H.c. \tag{15}$$

is composed only by highly oscillating terms, enabling, to a good approximation, an effective interaction [18, 19]

$$H_{eff} \approx -i\mathcal{H}(t) \int_0^t \mathcal{H}(t) d\tau \approx -\frac{|\lambda|^2}{\Delta} a^\dagger a \sigma_{gg} - \frac{\lambda^* \Omega_1}{\Delta_1} a^\dagger \sigma_{ge} e^{i(\Delta_1 - \Delta)t}$$

$$- \frac{\lambda \Omega_1^*}{\Delta} a \sigma_{eg} e^{-i(\Delta_1 - \Delta)t} + \sum_\ell (-1)^{\delta_{\ell 1}} \frac{|\Omega_\ell|^2}{\Delta_\ell} \sigma_{ee}.$$

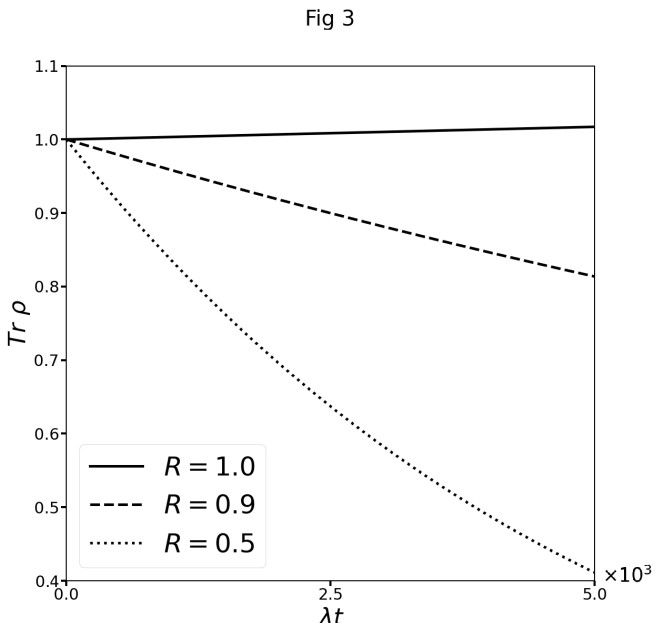

Figure 3: Plot of $\text{Tr}\,\rho(t)$ against $\lambda t$, for $R = 1.0$, 0.9, and 0.5 as indicated by straight, dashed and dot lines, respectively. We have considered $\alpha = 0.1$ and $\Delta_2 = \Delta_3 = 5 \times 10^3 \lambda$ for all values of $R$. However, for $R = 1$ we fixed (in units of $\lambda$) $\beta = 0.1$, $|\Omega_1| = 9.2$, $|\Omega_2| = |\Omega_3| = 48$ and $\Delta = \Delta_1 = 92$. for $R = 0.9$ we fixed $\beta = 0.09$, $|\Omega_1| = 6.6$, $|\Omega_2| = |\Omega_3| = 141.2$, $\Delta = 66.4$ and $\Delta_1 = 73.8$. Finally, for $R = 0.5$ we fixed $\beta = 0.05$, $|\Omega_1| = 10$, $|\Omega_2| = |\Omega_3| = 501.3$, $\Delta = 100$ and $\Delta_1 = 200$.

After a unitary transformation using $U(t) = e^{-i\chi \sigma_{ee} t}$, with $\chi = \sum_\ell (-1)^{\delta_{\ell 1}} |\Omega_\ell|^2 / \Delta_\ell = \Delta_1 - \Delta > 0$, we finally obtain, for $\beta \gg \bar{n}|\lambda|/\Delta$, the non-Hermitian effective interaction

$$H_{eff} \approx |\lambda|\,\alpha\left(a\sigma_+ + Ra^\dagger \sigma_-\right),\tag{16}$$

with $\alpha = |\Omega_1|/\Delta$ and $\beta = |\Omega_1|/\Delta_1$. The non-Hermiticity thus follows from the gap $\Delta_1 - \Delta = \chi$ which evidently increases with the number of pumping fields; and that is why we left this number arbitrary in our scheme. However, as already pointed out, even with an arbitrary number of pumping fields, our adiabatic elimination scheme is not efficient for the construction of far-from-Hermitian interactions, with $R \le 0.17$, since the pumping amplitudes must be limited by their detunings with the cavity mode.

We stress that although we started from a Hermitian Hamiltonian, the non-Hermiticity results from a second-order approximation in which $\mathcal{H}(t)$ does not in general commute with $\int_0^t \mathcal{H}(\tau)d\tau$. In short, for the regime of parameters we have considered, the originally Hermitian Hamiltonian $H$ reduces to the non-Hermitian second-order approximation $H_{eff}$. Indeed we verify that the norm of $H$ is no longer conserved under the parameters leading to $H_{eff}$, indicating that it actually becomes a non-Hermitian operator. In Fig. 3 we consider the evolution of $H$ to plot $\text{Tr}\,\rho(t)$ against $\lambda t$, $\rho(t)$ being the evolved atom-field density operator. We start with the field in the vacuum and the atom in the excited state, assuming the parameters given in the caption. The straight, dashed and dot lines refer to $R = 1$, 0.9, and 0.5, respectively, indicating that the norm decreases monotonically for $R = 0.9$ and 0.5. The small deviation from the unit observed for $R = 1$, follows from numerical errors.

In Fig. 4 we plot the mean excitation $\langle a^\dagger a\rangle(t)$ against $|\lambda|\,t$ computed from the full Hamiltonian (15) (dotted line) and the effective one (16) (solid line), starting again with the field

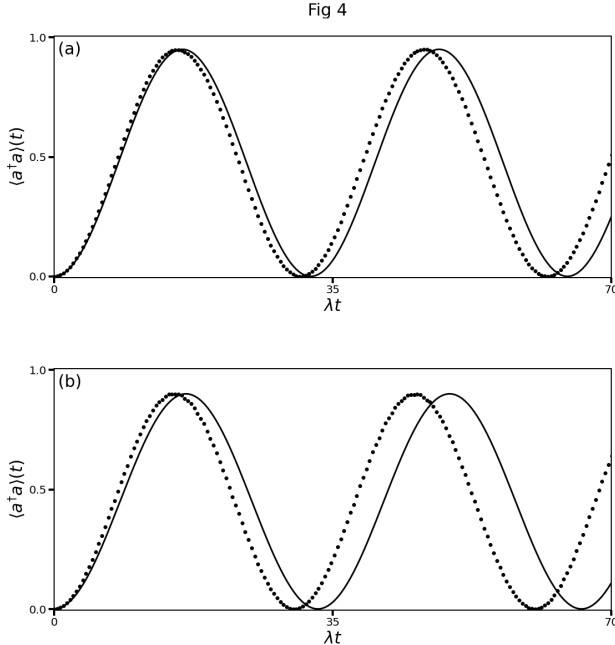

Figure 4: Plot of $\left\langle a^\dagger a \right\rangle(t)$ against $|\lambda| t$ computed from the full Hamiltonian (15) (dotted line) and the effective one (16) (solid line), for $R = 0.95$, and 0.9, considering $\alpha = 0.1$ and $\Delta_2 = \Delta_3 = 5 \times 10^3 \lambda$ for all values of $R$. For $R = 0.95$ we fixed (in units of $\lambda$) $\beta = 9.5 \times 10^{-2}$, $|\Omega_1| = 9.3$, $|\Omega_2| = |\Omega_3| = 120$, $\Delta = 92.6$ and $\Delta_1 = 97.5$. For $R = 0.9$ we fixed $\beta = 9 \times 10^{-2}$, $|\Omega_1| = 6.6$, $|\Omega_2| = |\Omega_3| = 141.2$, $\Delta = 66.4$ and $\Delta_1 = 73.8$.

in the vacuum, the atom in the excited state and considering the parameters given in the caption. Here we do not consider the metric $\Theta = \eta^\dagger \eta$ to compute the mean value $\left\langle a^\dagger a \right\rangle(t)$ for the non-Hermitian $H_{eff}$, i.e., we do not follow the prescription in Eq. (4), since we only seek to compare the dynamics generated by both Hamiltonians, without worrying about norm conservation. In Figs. 4 (a and b) we consider $R = 0.95$ and 0.9, respectively, to observe that for $R = 0.95$ the effective interaction is a good approximation of the full Hamilton for $|\lambda| t$ up to around 35. However, when we go to $R = 0.9$, the curves show discrepancies already for $|\lambda| t \approx 15$. In both cases the discrepancies are more pronounced in phases than in amplitudes, and increase as we move further away from Hermiticity, decreasing $R$.

While the engineering enabling $H \to H_{eff}$ follows from the adiabatic elimination method and the map $H_{eff} \to h$ follows from the pseudo-Hermiticity relation, both the adiabatic elimination and the pseudo-Hermiticity must be put together through the energy balance between $H$ and $h$. The impossibility of such a balance leads us to conclude that another engineering scheme must be developed in which the amplitudes of the pumping fields are not limited by their detunings with the mode.

## 4 Conclusion

The method here proposed for strengthening the Rabi coupling through pseudo-Hermitian Hamiltonians is similar to those for reaching infinite squeezing degree at finite times [20], for the enhancement of Casimir's photon creation [21], and for the strengthening of the Dicke superradiance intensity [22]. All these achievements rely on the engineering of interactions

which are far from Hermiticity, a challenge that remains to be accomplished. We stress that the non-Hermitian Hamiltonian (2) as well as those introduced in Ref. [20–22], must necessarily be engineered as effective Hamiltonians, since they result in (or leads to) energy gain, which must be provided by high amplitude amplification fields.

It is crucial to underline that the Dyson map $\eta$ in Eq. (5), used to ensure the pseudo-Hermiticity relation (and therefore the conservation of the norm in a new metric $\Theta = \eta^{\dagger}\eta$), basically implies new observables $\mathcal{O} = \eta^{-1}o\eta$ [7, 12] and consequently in the implementation of procedures for measuring canonically conjugated variables [16, 17]. Therefore, we stress that the present protocol is perfectly feasible once the engineering of the non-Hermitian Hamiltonian (2) is implemented, which is the really sensitive point for its pratical realization.

We have also discussed the energy cost for the remarkable gain in the atom-field coupling, which must be supported by the construction of the non-Hermitian Hamiltonian and by carrying out the measurements of canonically conjugated variables. We finally observe that, apart from the prospects for the implementation of the present method in platforms of radiation-matter interaction, we cannot but speculate on the impacts that the possible adaptation of the present method would bring to the field of high-energy experimental physics.

# Acknowledgments

**Funding information**    FSL and MHYM would like to thank CAPES and CNPQ, Brazilian agencies, for grants 88882.328726/2019-01 and 303012/2020-0, respectively.

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
