# Peer review of "Strengthening the atom-field coupling through the deep-strong regime via pseudo-Hermitian Hamiltonians"

_SciPost Physics, doi:SciPost Phys. 15, 091 (2023)_

## Round 1 · Referee Report · Anonymous (Referee 1) · 2023-1-20

Strengths

1/ The paper addresses a strategy to build a pseudo-Hermitian Hamiltonian with enhanced control of atom-field interactions. In general, doing so is not easy because the utility of the method, expressed by (4), relies on obtaining the Dyson map.

2/ An explicit expression for the Dyson map $\eta$ is proposed, and the requirements to ensure a Hermitian $h$ for each parameter are clearly outlined.

3/ The authors also explain in details how the non-Hermitian Hamiltonian is obtained in first place, making the paper self-contained and far more readable.

Weaknesses

1/ It was not clear to me how the authors arrived at their expression for the $\eta$ in (5). A brief explanation, followed by the appropriate reference would greatly help the reader to understand the motivation behind this particular choice for $\eta$.

2/ Some assumptions are made but poorly justified. For instance, just before (7), by assuming $\varphi_\beta +\varphi_\alpha = 0$.

3/ Although the figures are explained in the text, the lack of captions make it significantly harder.

Report

The authors explored the framework of pseudo-Hermitian Hamiltonians to improve the coupling between radiation and matter fields. The advantages and disadvantages of the method were discussed in a clear and simple manner.

The authors started by showing how to obtain a simple Hermitian operators starting from a non-Hermitian one, with the basic ingredients of the Jaynes-Cummings model. Then, they explain the effect of the Dyson map $\eta$ and the requirements to ensure $h$ remains real. Finally, the authors explain the steps to derive the pseudo-Hermitian Hamiltonian.

The paper is well written and I could not detect any error. For practical purposes, the paper reaches the standards for publication in SciPost with minimal changes .

Requested changes

1 / Clarify, even if briefly, the motivation to propose the Dyson map (5)

2/ Include verifications on the hypothesis, whether they have been upheld or not

3 / I got a bit lost at first because it was not explained why $H_{\textrm{eff}}$ would have the form (2). It only became clear around the middle of section III

4 / Please consider using captions for figures. It also odd to have a single subsection in section III.

5 / The authors should profit from the broader audience of SciPost and expand a bit on possible applications in other areas.

  • validity: high
  • significance: good
  • originality: good
  • clarity: high
  • formatting: good
  • grammar: excellent

Author:  Miled Moussa  on 2023-06-22  [id 3750]

(in reply to Report 1 on 2023-01-20)

To the Editor-in-charge of our submitted manuscript, entitled "Strengthening the atom-field coupling through the deep-strong regime via pseudo-Hermitian Hamiltonians", authored by M.A. de Ponte, F. de Oliveira Neto, and M.H.Y. Moussa.

Dear Editor,

Thank you for your message of Jun 21, regarding our submitted manuscript. We also thank the anonymous referee for his constructive Report 1 on 2023-01-20, which helped us to clarify the new version of the manuscript. Below we present our answeres to the anonymous referee.

    We begin by addressing the first weakness of the manuscript pointed out by anonymous referee:

1) It was not clear to me how the authors arrived at their expression for the η in (5). A brief explanation, followed by the appropriate reference would greatly help the reader to understand the motivation behind this particular choice for η.

The Dyson map in Eq. (5) is just an appropriate ansatz for the problem. We would not be able to obtain the results presented in the manuscript considering for example a linear Dyson map, of the type of a displacement operator for the radiation field. Regarding the atomic subspace, it was not necessary to consider anything other than the identity operator.

Therefore, as usual, we made an ansatz for the Dyson map, and then we calculated the parameters involved in it (as established in Eq. (7)), in order to ensure the validity of the pseudo-Hermiticity relation. In the new version of the manuscript we have now rewritten the paragraph above Eq. (5), pointing out that our Dyson map was an ansatz for the hermitization of the Jaynes-Cummings non-Hermitian Hamiltonian (2): "We next outline our protocol starting from the engineered Hamiltonian H_{eff} to construct its pseudo-Hermitian counterpart h through the ansatz for the nonunitary Dyson map...".

We also observe that we do not present any reference about the Dyson map as we know of no other work about the hermitization of the non-Hermitian Jaynes-Cummimgs Hamiltonian.

More importantly, we observe that, for a proper choice of the initial state of the system, it is not necessary to engineer a Dyson map in the laboratory. All that the Dyson map defines are the new observables, establishing the new metric of the problem. These new observables, as in Eq. (12), thus impose the task of measuring superpositions of canonically conjugate variables, a problem that has long been addressed in the literature as indicated by the Refs. [16,17].

The second weakness of the manuscript pointed out by anonymous referee, is:
    2) Some assumptions are made but poorly justified. For instance, just before (7), by assuming ϕ_{β}+ϕ_{α}=0.

The assumption that ϕ_{β}+ϕ_{α}=0, must be fulfilled when engineering the non-Hermitian Hamiltonian defined in Eq. (2). The parameters alpha and beta that appear in this equation must consider that ϕ_{β}+ϕ_{α}=0. In the new version of the manuscript we have now rewritten the paragraph above Eq. (7) to: "Assuming α=|α|e^{iϕ_{α}} and β=|β|e^{iϕ_{β}} with ϕ_{β}=-ϕ_{α}, a condition that must be imposed when engineering the Hamiltonian <ref>H</ref>, the Hermiticity of h demands the relations...".

The third weakness of the manuscript pointed out by anonymous referee, is:

3) Although the figures are explained in the text, the lack of captions make it significantly harder.

On this regards, we note that Caption Figures were provided in the manuscript.

The anonymous referee has also requested some changes in the manuscript, starting with: the clarification of "the motivation to propose the Dyson map (5)". We believe that this point has been properly addressed in item 1) of our response to the anonymous referee: the Dyson map we used was an anstatz that proved to be suitable for our problem.

The anonymous referee also requested us to "include verifications on the hypothesis, whether they have been upheld or not". We are not sure that we fully understand this question raised by the referee. In our manuscript we have basically considered a non-Hermitian Hamiltonian and used the Mostafazadeh's method to derive the experimentally verifiable physical consequences that follow from it. These physical consequences result from the new metric we have used, which stems from the Dyson map in the form Θ=η^{†}η. This new metric translates into the new observables from which we define the physical quantities associated with the system described by our Hamiltonian. All that our proposal requires for its implementation is the engineering of the effective non-Hermitian Hamiltonian (2), which together with the measurements of canonically conjugated variables, are the main challenges of our proposal as we clarify in the conclusions. Therefore, considering that the measurement of canonically conjugated variables is something already established in the literature, the only hypothesis that we made in our manuscript, so that we can experimentally prove the feasibility of our method, is that of the possibility of engineering Hamiltonians sufficiently far from hermiticity.

The anonymous referee also observe that "I got a bit lost at first because it was not explained why H_{eff} would have the form (2). It only became clear around the middle of section III". On this regard we just note that the introduction of the parameters α and β into both terms of the Jaynes-Cummings Hamiltonian is just the simplest way to make it non-Hermitian.

Then, the anonymous referee requests that we use Caption Figures in our manuscript, which we have already done. And finally, the referee observes that "The authors should profit from the broader audience of SciPost and expand a bit on possible applications in other areas." This point raised by the anonymous referee is really essential. In the conclusions, we just limit ourselves to writing that "apart from the prospects for the implementation of the present method in platforms of radiation-matter interaction, we cannot but speculate on the impacts that the possible adaptation of the present method would bring to the field of high-energy experimental physics." Maybe our method can be used in high energy physics, but It is still premature to make any statement about this. For this reason, we decided to better analyze the topic before anticipating possible other applications beyond the field of radiation-matter interaction.

We again thank the anonymous referees for the positive criticisms that helped us to improve our manuscript, making its new version clearer and more complete.

Best regards,
Flavio de Oliveira Neto, Mickel de Ponte and Miled Moussa.

Attachment:

G-Grande-Novo-130.pdf

---

## Round 1 · Referee Report · Shunyu Yao (Referee 2) · 2023-5-21

Report

The authors proposed a way to enhance atom-field coupling via non-hermitian generalization of Jaynes-Cummings model. The technical details are valid and the idea of using non-hermitian systems to enhance quantum informational process worth further pursue. Thus I recommended this paper to be published.

However I strongly recommend the following few modifications the authors should consider, in order to improve the readability of the paper.

1.The author discussed construction of pseudo-hermitian Hamiltonian, following ref.7. These are surely correct, however, if I understand correctly, a easier way to conclude Eq.(3)-Eq.(4) and discussion related to those, is just saying you construct your Hamiltonian $H_{eff}$, such that it equal to an hermitian Hamiltonian $h$ via a similarity transformation $\eta$. And the inner product defined in Eq.(4), is just $\langle \Psi(t)|O|\Psi(t)\rangle$.

(And the right eigenstate for $H_{eff}$ are related to eigenstate of $h$ via $|\Psi_{n,R}\rangle=\eta^{-1} |\psi_n\rangle$, while left eigenvector is $\langle\Psi_{n,L}|= \langle\psi_n|\eta$. Actually these are more widely used way of describing non-hermitian systems, based on development of recent years. )

  1. I would like the authors to clarify Eq.(12a,12b), where the operator has a large coefficient $\mathcal{A}$ and $\mathcal{B}$, in the untransformed basis. I understand this does not necessarily cause a problem, because that coefficients are not necessarily connects to physical quantities, but can the author comments more on how experimentally feasible for measuring this operator? Does measuring this operator cost a high energy?
  • validity: good
  • significance: ok
  • originality: good
  • clarity: low
  • formatting: good
  • grammar: excellent

Author:  Miled Moussa  on 2023-06-22  [id 3751]

(in reply to Report 2 by Shunyu Yao on 2023-05-21)

To the Editor-in-charge of our submitted manuscript, entitled "Strengthening the atom-field coupling through the deep-strong regime via pseudo-Hermitian Hamiltonians", authored by M.A. de Ponte, F. de Oliveira Neto, and M.H.Y. Moussa.

Dear Editor,

Thank you for your message of May 21, regarding our submitted manuscript. We also thank Dr. Yao for his constructive comments that helped us eliminate an incorrect statement that was in the first version of the manuscript. Below we present our answers to Dr. Yao.

The first comment made by Dr. Yao is the following:

1.The author discussed construction of pseudo-Hermitian Hamiltonian, following ref.7. These are surely correct, however, if I understand correctly, a easier way to conclude Eq.(3)-Eq.(4) and discussion related to those, is just saying you construct your Hamiltonian H_{eff}, such that it equal to an Hermitian Hamiltonian h via a similarity transformation η. And the inner product defined in Eq.(4), is just <Ψ(t)|O|Ψ(t)>.
(And the right eigenstate for H_{eff}, are related to eigenstate of h via |Ψ_{n,R}>=η⁻¹|ψ_{n}>, while left eigenvector is <Ψ_{n,R}|=<ψ_{n}|η. Actually these are more widely used way of describing non-Hermitian systems, based on development of recent years.)

We fully agree with this observation. We can indeed construct the effective Hamiltonian H_{eff}, such that it equal to a Hermitian Hamiltonian h via a similarity transformation η, where the non-unitary operator η defines a new metric Θ=η^{†}η, through which we redefine the inner product. In our manuscript, however, we chose to closely follow the usual form presented by Mostafazadeh in Ref. [7].

The second comment made by Dr. Yao is:

2. I would like the authors to clarify Eq.(12a,12b), where the operator has a large coefficient A and B, in the untransformed basis. I understand this does not necessarily cause a problem, because that coefficients are not necessarily connects to physical quantities, but can the author comments more on how experimentally feasible for measuring this operator? Does measuring this operator cost a high energy?

The simultaneous measurements of canonically conjugated variables seem to be perfectly feasible in conventional quantum optics. As noted by U. Leonhardt and H Paul, in Ref. [14], these measurements are supported by "well-known schemes (...) based on beam-splitting, amplification and heterodyning." These measurements are really a sensitive issue in our scheme, and in the manuscript we limit ourselves to drawing attention to the fact. Equally difficult as engineering a Hamiltonian with a significantly small degree of Hermiticity is to carry out these measurements of canonically conjugated variables. We hope that these difficulties have been properly highlighted in the manuscript.

Regarding the other question very appropriately raised by the referee: "Does measuring this operator cost a high energy?", we note that we have completely modified the discussion on this topic presented in the manuscript. In fact, we cannot make the claim found in the manuscript: "The higher the Rabi frequencies, the higher the energies required for measuring properties of the radiation field, as higher as the lower the error tolerances." For an in-depth discussion of this topic, we refer to the well-known works on the relationship between energy-time uncertainty and quantum measurements:

V. Fock and N. Krylov, J. Phys., USSR, 11, 112 (1947);
Y. Aharonov and D. Bohm, Phys. Rev. 122, 1649 (1961);
V.A. Fock, Sov. Phys. - JETP 15,784 (1962);
Y. Aharonov and D. Bohm, Phys. Rev. 134B, 1417 (1964);
Y. Aharonov and J. L. Sajko, Ann. Phys. (N.Y.) 91, 279 (1975);
Yu. I. Vorontsov, Sov. Phys. - Uspekhi 24, 150 (1981);
M. Moshinsky, Am. J.Phys. 44, 1037 (1976);
J. Rayski and J. M. Rayski, Jr., "On the meaning of the time-energy uncertainty relation", in The Uncertainty Relation and Foundations of
Quantum Mechanic, edited by W. C. Price and S. S. Chissick (John Wiley, New York, 1971).

In the new version of the manuscript we have modified the sentence correctly questioned by the referee, replacing it with the paragraph:

"Regarding achieving faster than Hermitian quantum mechanics, we note that the effective coupling strength G defines a typical time 1/G to carry out an elementary logical operation. The minimum energy required for this operation, over a given error tolerance ε, is estimated to be E_{min}≈ℏG/ε [13]. The higher the Rabi frequencies, the higher the energies required for this fast than Hermitian quantum operation, as higher as the lower the error tolerances."
"We mention here a recently presented result [14], where it is demonstrated that the construction of coherent many-body Rabi oscillations, through the coherent interaction of an atomic sample with a field mode, allows increasing the Rabi frequency g by the factor √N, where N is the number of atoms in the sample. In this case, the typical time to carry out an elementary logical operation decreases from 1/g to 1/√Ng. Therefore, in addition to the gain in computational time that results from the quantum nature of the operation, i.e., from the use of qubits as information carriers [15], we have here the gain that results from the collective nature of the radiation-matter interaction. In the present proposal, the gain in computational time comes from strengthening the Rabi frequency through pseudo-Hermiticity instead of taking advantage of collective effects in the coherent interaction between atomic samples and cavity fields."

    With the changes made to the new version of the manuscript, resulting from the observations by Dr. Yao, we made the manuscript clearer and now more complete.

Best regards,
Flavio de Oliveira Neto, Mickel de Ponte and Miled Moussa.

Attachment:

G-Grande-Novo-130_cBu5Vb5.pdf

Author:  Miled Moussa  on 2023-06-21  [id 3747]

(in reply to Report 2 by Shunyu Yao on 2023-05-21)

To the Editor-in-charge of our submitted manuscript, entitled "Strengthening the atom-field coupling through the deep-strong regime via pseudo-Hermitian Hamiltonians", authored by M.A. de Ponte, F. de Oliveira Neto, and M.H.Y. Moussa.

Dear Editor,

Thank you for your message of Jun 21, regarding our submitted manuscript. We also thank Dr. Yao for his constructive Report 2 on 2023-05-21, that helped us eliminate an incorrect statement that was in the first version of the manuscript. Finally we thank the Anonymous referee for his equally constructive Report 1 on 2023-01-20. Below we first present our responses to Dr. Yao, and then to the Anonymous referee (Report 1).

Our answers to Report 2, by Dr. Yao:

The first comment made by Dr. Yao is the following:

1.The author discussed construction of pseudo-Hermitian Hamiltonian, following ref.7. These are surely correct, however, if I understand correctly, a easier way to conclude Eq.(3)-Eq.(4) and discussion related to those, is just saying you construct your Hamiltonian H_{eff}, such that it equal to an Hermitian Hamiltonian h via a similarity transformation η. And the inner product defined in Eq.(4), is just <Ψ(t)|O|Ψ(t)>.
(And the right eigenstate for H_{eff}, are related to eigenstate of h via |Ψ_{n,R}>=η⁻¹|ψ_{n}>, while left eigenvector is <Ψ_{n,R}|=<ψ_{n}|η. Actually these are more widely used way of describing non-Hermitian systems, based on development of recent years.)

We fully agree with this observation. We can indeed construct the effective Hamiltonian H_{eff}, such that it equal to a Hermitian Hamiltonian h via a similarity transformation η, where the non-unitary operator η defines a new metric Θ=η^{†}η, through which we redefine the inner product. In our manuscript, however, we chose to closely follow the usual form presented by Mostafazadeh in Ref. [7].

The second comment made by Dr. Yao is:

2. I would like the authors to clarify Eq.(12a,12b), where the operator has a large coefficient A and B, in the untransformed basis. I understand this does not necessarily cause a problem, because that coefficients are not necessarily connects to physical quantities, but can the author comments more on how experimentally feasible for measuring this operator? Does measuring this operator cost a high energy?

The simultaneous measurements of canonically conjugated variables seem to be perfectly feasible in conventional quantum optics. As noted by U. Leonhardt and H Paul, in Ref. [14], these measurements are supported by "well-known schemes (...) based on beam-splitting, amplification and heterodyning." These measurements are really a sensitive issue in our scheme, and in the manuscript we limit ourselves to drawing attention to the fact. Equally difficult as engineering a Hamiltonian with a significantly small degree of Hermiticity is to carry out these measurements of canonically conjugated variables. We hope that these difficulties have been properly highlighted in the manuscript.

Regarding the other question very appropriately raised by the referee: "Does measuring this operator cost a high energy?", we note that we have completely modified the discussion on this topic presented in the manuscript. In fact, we cannot make the claim found in the manuscript: "The higher the Rabi frequencies, the higher the energies required for measuring properties of the radiation field, as higher as the lower the error tolerances." For an in-depth discussion of this topic, we refer to the well-known works on the relationship between energy-time uncertainty and quantum measurements:

V. Fock and N. Krylov, J. Phys., USSR, 11, 112 (1947);
Y. Aharonov and D. Bohm, Phys. Rev. 122, 1649 (1961);
V.A. Fock, Sov. Phys. - JETP 15,784 (1962);
Y. Aharonov and D. Bohm, Phys. Rev. 134B, 1417 (1964);
Y. Aharonov and J. L. Sajko, Ann. Phys. (N.Y.) 91, 279 (1975);
Yu. I. Vorontsov, Sov. Phys. - Uspekhi 24, 150 (1981);
M. Moshinsky, Am. J.Phys. 44, 1037 (1976);
J. Rayski and J. M. Rayski, Jr., "On the meaning of the time-energy uncertainty relation", in The Uncertainty Relation and Foundations of
Quantum Mechanic, edited by W. C. Price and S. S. Chissick (John Wiley, New York, 1971).

In the new version of the manuscript we have modified the sentence correctly questioned by the referee, replacing it with the paragraph:

"Regarding achieving faster than Hermitian quantum mechanics, we note that the effective coupling strength G defines a typical time 1/G to carry out an elementary logical operation. The minimum energy required for this operation, over a given error tolerance ε, is estimated to be E_{min}≈ℏG/ε [13]. The higher the Rabi frequencies, the higher the energies required for this fast than Hermitian quantum operation, as higher as the lower the error tolerances."
"We mention here a recently presented result [14], where it is demonstrated that the construction of coherent many-body Rabi oscillations, through the coherent interaction of an atomic sample with a field mode, allows increasing the Rabi frequency g by the factor √N, where N is the number of atoms in the sample. In this case, the typical time to carry out an elementary logical operation decreases from 1/g to 1/√Ng. Therefore, in addition to the gain in computational time that results from the quantum nature of the operation, i.e., from the use of qubits as information carriers [15], we have here the gain that results from the collective nature of the radiation-matter interaction. In the present proposal, the gain in computational time comes from strengthening the Rabi frequency through pseudo-Hermiticity instead of taking advantage of collective effects in the coherent interaction between atomic samples and cavity fields."

Our answers to Report 1, by the anonymous referee:

We begin by addressing the first weakness of the manuscript pointed out by anonymous referee:

1) It was not clear to me how the authors arrived at their expression for the η in (5). A brief explanation, followed by the appropriate reference would greatly help the reader to understand the motivation behind this particular choice for η.

The Dyson map in Eq. (5) is just an appropriate ansatz for the problem. We would not be able to obtain the results presented in the manuscript considering for example a linear Dyson map, of the type of a displacement operator for the radiation field. Regarding the atomic subspace, it was not necessary to consider anything other than the identity operator.

Therefore, as usual, we made an ansatz for the Dyson map, and then we calculated the parameters involved in it (as established in Eq. (7)), in order to ensure the validity of the pseudo-Hermiticity relation. In the new version of the manuscript we have now rewritten the paragraph above Eq. (5), pointing out that our Dyson map was an ansatz for the hermitization of the Jaynes-Cummings non-Hermitian Hamiltonian (2): "We next outline our protocol starting from the engineered Hamiltonian H_{eff} to construct its pseudo-Hermitian counterpart h through the ansatz for the nonunitary Dyson map...".

We also observe that we do not present any reference about the Dyson map as we know of no other work about the hermitization of the non-Hermitian Jaynes-Cummimgs Hamiltonian.

More importantly, we observe that, for a proper choice of the initial state of the system, it is not necessary to engineer a Dyson map in the laboratory. All that the Dyson map defines are the new observables, establishing the new metric of the problem. These new observables, as in Eq. (12), thus impose the task of measuring superpositions of canonically conjugate variables, a problem that has long been addressed in the literature as indicated by the Refs. [16,17].

The second weakness of the manuscript pointed out by anonymous referee, is:

    2) Some assumptions are made but poorly justified. For instance, just before (7), by assuming φ_{β}+φ_{α}=0.

The assumption that φ_{β}+φ_{α}=0, must be fulfilled when engineering the non-Hermitian Hamiltonian defined in Eq. (2). The parameters alpha and beta that appear in this equation must consider that φ_{β}+φ_{α}=0. In the new version of the manuscript we have now rewritten the paragraph above Eq. (7) to: "Assuming α=|α|e^{iϕ_{α}} and β=|β|e^{iϕ_{β}} with ϕ_{β}=-ϕ_{α}, a condition that must be imposed when engineering the Hamiltonian <ref>H</ref>, the Hermiticity of h demands the relations...".

The third weakness of the manuscript pointed out by anonymous referee, is:

3) Although the figures are explained in the text, the lack of captions make it significantly harder.

On this regards, we note that Caption Figures were provided in the manuscript.

The anonymous referee has also requested some changes in the manuscript, starting with: the clarification of "the motivation to propose the Dyson map (5)". We believe that this point has been properly addressed in item 1) of our response to the anonymous referee: the Dyson map we used was an anstatz that proved to be suitable for our problem.

The anonymous referee also requested us to "include verifications on the hypothesis, whether they have been upheld or not". On this regard we note that the Mostafazadeh's method (Ref. [7]) used in our manuscript is standard for aaproaching non-Hermitian Hamiltonians. Therefore, all that our proposal requires for its implementation is the engineering of the effective non-Hermitian Hamiltonian (2), which together with the measurements of canonically conjugated variables, are the main challenges of our proposal as we clarify in the conclusions.

    The anonymous referee also observe that "I got a bit lost at first because it was not explained why H_{eff} would have the form (2). It only became clear around the middle of section III". We observe that the introduction of the parameters α and β into both terms of the Jaynes-Cummings Hamiltonian is just the simplest way to make it non-Hermitian.

    Then, the anonymous referee requests that we use Caption Figures in our manuscript, which we have already done. And finally, the referee observes that "The authors should profit from the broader audience of SciPost and expand a bit on possible applications in other areas." This point raised by the anonymous referee is really essential. In the conclusions, we just limit ourselves to writing that "apart from the prospects for the implementation of the present method in platforms of radiation-matter interaction, we cannot but speculate on the impacts that the possible adaptation of the present method would bring to the field of high-energy experimental physics." Maybe our method can be used in high energy physics, but It is still premature to make any statement about this. For this reason, we decided to better analyze the topic before anticipating possible other applications beyond the field of radiation-matter interaction.

We again thank the referees for the positive criticisms that helped us to improve our manuscript, making its new version clearer and more complete.

Best regards,
Flavio de Oliveira Neto, Mickel de Ponte and Miled Moussa.

Attachment:

G-Grande-Novo-130.pdf

---

## Round 2 · Author Response

To the Editor-in-charge of our submitted manuscript, entitled "Strengthening
the atom-field coupling through the deep-strong regime via pseudo-Hermitian
Hamiltonians", authored by M.A. de Ponte, F. de Oliveira Neto, and M.H.Y.
Moussa.
Dear Editor,
Thank you for your message of May 21, regarding our submitted manuscript. We
also thank Dr. Yao for his constructive comments that helped us eliminate an
incorrect statement that was in the first version of the manuscript. Below
we present our answers to Dr. Yao.
The first comment made by Dr. Yao is the following:
\textbf{1.The author discussed construction of pseudo-hermitian Hamiltonian,
following ref.7. These are surely correct, however, if I understand
correctly, a easier way to conclude Eq.(3)-Eq.(4) and discussion related to
those, is just saying you construct your Hamiltonian }$H_{eff}$\textbf{,
such that it equal to an hermitian Hamiltonian }$h$\textbf{\ via a
similarity transformation }$\eta $\textbf{. And the inner product defined in
Eq.(4), is just }$\left\langle \Psi (t)\left\vert O\right\vert \Psi
(t)\right\rangle $\textbf{.}
\textbf{(And the right eigenstate for }$H_{eff}$\textbf{, are related to
eigenstate of }$h$\textbf{\ via }$\left\vert \Psi _{n,R}\right\rangle =\eta
^{-1}\left\vert \psi _{n}\right\rangle $\textbf{, while left eigenvector is }%
$\left\langle \Psi _{n,R}\right\vert =\left\langle \psi _{n}\right\vert \eta
$\textbf{. Actually these are more widely used way of describing
non-hermitian systems, based on development of recent years.)}
We fully agree with this observation. We can indeed construct the effective
Hamiltonian $H_{eff}$, such that it equal to a Hermitian Hamiltonian $h$ via
a similarity transformation $\eta $, where the non-unitary operator $\eta $
defines a new metric $\Theta =\eta ^{\dag }\eta $, through which we redefine
the inner product. In our manuscript, however, we chose to closely follow
the usual form presented by Mostafazadeh in Ref. [7].
The second comment made by Dr. Yao is:
\textbf{2. I would like the authors to clarify Eq.(12a,12b), where the
operator has a large coefficient }$A$\textbf{\ and }$B$\textbf{, in the
untransformed basis. I understand this does not necessarily cause a problem,
because that coefficients are not necessarily connects to physical
quantities, but can the author comments more on how experimentally feasible
for measuring this operator? Does measuring this operator cost a high energy?%
}\bigskip
The simultaneous measurements of canonically conjugated variables seem to be
perfectly feasible in conventional quantum optics. As noted by U. Leonhardt
and H Paul, in Ref. [14], these measurements are supported by "well-known
schemes (...) based on beam-splitting, amplification and heterodyning."
These measurements are really a sensitive issue in our scheme, and in the
manuscript we limit ourselves to drawing attention to the fact. Equally
difficult as engineering a Hamiltonian with a significantly small degree of
Hermiticity is to carry out these measurements of canonically conjugated
variables. We hope that these difficulties have been properly highlighted in
the manuscript.
Regarding the other question very appropriately raised by the referee: "Does
measuring this operator cost a high energy?", we note that we have
completely modified the discussion on this topic presented in the
manuscript. In fact, we cannot make the claim found in the manuscript: "The
higher the Rabi frequencies, the higher the energies required for measuring
properties of the radiation field, as higher as the lower the error
tolerances." For an in-depth discussion of this topic, we refer to the
well-known works on the relationship between energy-time uncertainty and
quantum measurements:
V. Fock and N. Krylov, J. Phys., USSR, 11, 112 (1947);
Y. Aharonov and D. Bohm, Phys. Rev. 122, 1649 (1961);
V.A. Fock, Sov. Phys. - JETP 15,784 (1962);
Y. Aharonov and D. Bohm, Phys. Rev. 134B, 1417 (1964);
Y. Aharonov and J. L. Sajko, Ann. Phys. (N.Y.) 91, 279 (1975);
Yu. 1. Vorontsov, Sov. Phys. - Uspekhi 24, 150 (1981);
M. Moshinsky, Am. J.Phys. 44, 1037 (1976);
J. Rayski and J. M. Rayski, Jr., "On the meaning of the time-energy
uncertainty relation", in The Uncertainty Relation and Foundations of
Quantum Mechanic, edited by W. C. Price and S. S. Chissick (John Wiley, New
York, 1971).\bigskip
In the new version of the manuscript we have modified the sentence correctly
questioned by the referee, replacing it with the paragraph:
\textquotedblleft Regarding achieving faster than Hermitian quantum
mechanics, we note that the effective coupling strength $G$ defines a
typical time $1/G$ to carry out an elementary logical operation. The minimum
energy required for this operation, over a given error tolerance $%
\varepsilon $, is estimated to be $E_{min}\approx \hbar G/\varepsilon $
[13]. The higher the Rabi frequencies, the higher the energies required for
this fast than Hermitian quantum operation, as higher as the lower the error
tolerances.\textquotedblright
\textquotedblleft We mention here a recently presented result [14], where it
is demonstrated that the construction of coherent many-body Rabi
oscillations, through the coherent interaction of an atomic sample with a
field mode, allows increasing the Rabi frequency $g$ by the factor $\sqrt{N}$%
, where $N$ is the number of atoms in the sample. In this case, the typical
time to carry out an elementary logical operation decreases from $1/g$ to $1/%
\sqrt{N}g$. Therefore, in addition to the gain in computational time that
results from the quantum nature of the operation, i.e., from the use of
qubits as information carriers [15], we have here the gain that results from
the collective nature of the radiation-matter interaction. In the present
proposal, the gain in computational time comes from strengthening the Rabi
frequency through pseudo-Hermiticity instead of taking advantage of
collective effects in the coherent interaction between atomic samples and
cavity fields.\textquotedblright
With the changes made to the new version of the manuscript, resulting from
the observations by Dr. Yao, we made the manuscript clearer and now more
complete.
Best regards,
Flavio de Oliveira Neto, Mickel de Ponte and Miled Moussa.
the atom-field coupling through the deep-strong regime via pseudo-Hermitian
Hamiltonians", authored by M.A. de Ponte, F. de Oliveira Neto, and M.H.Y.
Moussa.
Dear Editor,
Thank you for your message of May 21, regarding our submitted manuscript. We
also thank Dr. Yao for his constructive comments that helped us eliminate an
incorrect statement that was in the first version of the manuscript. Below
we present our answers to Dr. Yao.
The first comment made by Dr. Yao is the following:
\textbf{1.The author discussed construction of pseudo-hermitian Hamiltonian,
following ref.7. These are surely correct, however, if I understand
correctly, a easier way to conclude Eq.(3)-Eq.(4) and discussion related to
those, is just saying you construct your Hamiltonian }$H_{eff}$\textbf{,
such that it equal to an hermitian Hamiltonian }$h$\textbf{\ via a
similarity transformation }$\eta $\textbf{. And the inner product defined in
Eq.(4), is just }$\left\langle \Psi (t)\left\vert O\right\vert \Psi
(t)\right\rangle $\textbf{.}
\textbf{(And the right eigenstate for }$H_{eff}$\textbf{, are related to
eigenstate of }$h$\textbf{\ via }$\left\vert \Psi _{n,R}\right\rangle =\eta
^{-1}\left\vert \psi _{n}\right\rangle $\textbf{, while left eigenvector is }%
$\left\langle \Psi _{n,R}\right\vert =\left\langle \psi _{n}\right\vert \eta
$\textbf{. Actually these are more widely used way of describing
non-hermitian systems, based on development of recent years.)}
We fully agree with this observation. We can indeed construct the effective
Hamiltonian $H_{eff}$, such that it equal to a Hermitian Hamiltonian $h$ via
a similarity transformation $\eta $, where the non-unitary operator $\eta $
defines a new metric $\Theta =\eta ^{\dag }\eta $, through which we redefine
the inner product. In our manuscript, however, we chose to closely follow
the usual form presented by Mostafazadeh in Ref. [7].
The second comment made by Dr. Yao is:
\textbf{2. I would like the authors to clarify Eq.(12a,12b), where the
operator has a large coefficient }$A$\textbf{\ and }$B$\textbf{, in the
untransformed basis. I understand this does not necessarily cause a problem,
because that coefficients are not necessarily connects to physical
quantities, but can the author comments more on how experimentally feasible
for measuring this operator? Does measuring this operator cost a high energy?%
}\bigskip
The simultaneous measurements of canonically conjugated variables seem to be
perfectly feasible in conventional quantum optics. As noted by U. Leonhardt
and H Paul, in Ref. [14], these measurements are supported by "well-known
schemes (...) based on beam-splitting, amplification and heterodyning."
These measurements are really a sensitive issue in our scheme, and in the
manuscript we limit ourselves to drawing attention to the fact. Equally
difficult as engineering a Hamiltonian with a significantly small degree of
Hermiticity is to carry out these measurements of canonically conjugated
variables. We hope that these difficulties have been properly highlighted in
the manuscript.
Regarding the other question very appropriately raised by the referee: "Does
measuring this operator cost a high energy?", we note that we have
completely modified the discussion on this topic presented in the
manuscript. In fact, we cannot make the claim found in the manuscript: "The
higher the Rabi frequencies, the higher the energies required for measuring
properties of the radiation field, as higher as the lower the error
tolerances." For an in-depth discussion of this topic, we refer to the
well-known works on the relationship between energy-time uncertainty and
quantum measurements:
V. Fock and N. Krylov, J. Phys., USSR, 11, 112 (1947);
Y. Aharonov and D. Bohm, Phys. Rev. 122, 1649 (1961);
V.A. Fock, Sov. Phys. - JETP 15,784 (1962);
Y. Aharonov and D. Bohm, Phys. Rev. 134B, 1417 (1964);
Y. Aharonov and J. L. Sajko, Ann. Phys. (N.Y.) 91, 279 (1975);
Yu. 1. Vorontsov, Sov. Phys. - Uspekhi 24, 150 (1981);
M. Moshinsky, Am. J.Phys. 44, 1037 (1976);
J. Rayski and J. M. Rayski, Jr., "On the meaning of the time-energy
uncertainty relation", in The Uncertainty Relation and Foundations of
Quantum Mechanic, edited by W. C. Price and S. S. Chissick (John Wiley, New
York, 1971).\bigskip
In the new version of the manuscript we have modified the sentence correctly
questioned by the referee, replacing it with the paragraph:
\textquotedblleft Regarding achieving faster than Hermitian quantum
mechanics, we note that the effective coupling strength $G$ defines a
typical time $1/G$ to carry out an elementary logical operation. The minimum
energy required for this operation, over a given error tolerance $%
\varepsilon $, is estimated to be $E_{min}\approx \hbar G/\varepsilon $
[13]. The higher the Rabi frequencies, the higher the energies required for
this fast than Hermitian quantum operation, as higher as the lower the error
tolerances.\textquotedblright
\textquotedblleft We mention here a recently presented result [14], where it
is demonstrated that the construction of coherent many-body Rabi
oscillations, through the coherent interaction of an atomic sample with a
field mode, allows increasing the Rabi frequency $g$ by the factor $\sqrt{N}$%
, where $N$ is the number of atoms in the sample. In this case, the typical
time to carry out an elementary logical operation decreases from $1/g$ to $1/%
\sqrt{N}g$. Therefore, in addition to the gain in computational time that
results from the quantum nature of the operation, i.e., from the use of
qubits as information carriers [15], we have here the gain that results from
the collective nature of the radiation-matter interaction. In the present
proposal, the gain in computational time comes from strengthening the Rabi
frequency through pseudo-Hermiticity instead of taking advantage of
collective effects in the coherent interaction between atomic samples and
cavity fields.\textquotedblright
With the changes made to the new version of the manuscript, resulting from
the observations by Dr. Yao, we made the manuscript clearer and now more
complete.
Best regards,
Flavio de Oliveira Neto, Mickel de Ponte and Miled Moussa.

---

## Round 2 · List of Changes

In the new version of the manuscript we have modified the sentence
"The higher the Rabi frequencies, the higher the energies required for measuring properties of the radiation field, as higher as the lower the error tolerances."
by
"Regarding achieving faster than Hermitian quantum
mechanics, we note that the effective coupling strength $G$ defines a
typical time $1/G$ to carry out an elementary logical operation. The minimum
energy required for this operation, over a given error tolerance $%
\varepsilon $, is estimated to be $E_{min}\approx \hbar G/\varepsilon $
[13]. The higher the Rabi frequencies, the higher the energies required for
this fast than Hermitian quantum operation, as higher as the lower the error
tolerances."
"We mention here a recently presented result [14], where it
is demonstrated that the construction of coherent many-body Rabi
oscillations, through the coherent interaction of an atomic sample with a
field mode, allows increasing the Rabi frequency $g$ by the factor $\sqrt{N}$%
, where $N$ is the number of atoms in the sample. In this case, the typical
time to carry out an elementary logical operation decreases from $1/g$ to $1/%
\sqrt{N}g$. Therefore, in addition to the gain in computational time that
results from the quantum nature of the operation, i.e., from the use of
qubits as information carriers [15], we have here the gain that results from
the collective nature of the radiation-matter interaction. In the present
proposal, the gain in computational time comes from strengthening the Rabi
frequency through pseudo-Hermiticity instead of taking advantage of
collective effects in the coherent interaction between atomic samples and
cavity fields."
"The higher the Rabi frequencies, the higher the energies required for measuring properties of the radiation field, as higher as the lower the error tolerances."
by
"Regarding achieving faster than Hermitian quantum
mechanics, we note that the effective coupling strength $G$ defines a
typical time $1/G$ to carry out an elementary logical operation. The minimum
energy required for this operation, over a given error tolerance $%
\varepsilon $, is estimated to be $E_{min}\approx \hbar G/\varepsilon $
[13]. The higher the Rabi frequencies, the higher the energies required for
this fast than Hermitian quantum operation, as higher as the lower the error
tolerances."
"We mention here a recently presented result [14], where it
is demonstrated that the construction of coherent many-body Rabi
oscillations, through the coherent interaction of an atomic sample with a
field mode, allows increasing the Rabi frequency $g$ by the factor $\sqrt{N}$%
, where $N$ is the number of atoms in the sample. In this case, the typical
time to carry out an elementary logical operation decreases from $1/g$ to $1/%
\sqrt{N}g$. Therefore, in addition to the gain in computational time that
results from the quantum nature of the operation, i.e., from the use of
qubits as information carriers [15], we have here the gain that results from
the collective nature of the radiation-matter interaction. In the present
proposal, the gain in computational time comes from strengthening the Rabi
frequency through pseudo-Hermiticity instead of taking advantage of
collective effects in the coherent interaction between atomic samples and
cavity fields."

---

## Editorial Decision

published